Plastics in Porifera: The occurrence of potential microplastics in marine sponges and seawater from Bocas del Toro, Panamá

Fallon Bailey R. fallonbr@g.cofc.edu
Freeman Christopher J.
Department of Biology, College of Charleston , Charleston, SC , United States
Pawlik Joseph
Electronic publication date: 2021 Jul 8
Publication date: 2021
Volume: 9
Electronic Location ID: e11638
Received 2021 Jan 6; Accepted 2021 May 28
Copyright: © 2021 Fallon and Freeman
Copyright year: 2021
Copyright holder: Fallon and Freeman
License: This is an open access article distributed under the terms of the Creative Commons Attribution License, which permits unrestricted use, distribution, reproduction and adaptation in any medium and for any purpose provided that it is properly attributed. For attribution, the original author(s), title, publication source (PeerJ) and either DOI or URL of the article must be cited.
License URL: https://creativecommons.org/licenses/by/4.0/

Keywords: Anthropogenic pollution; Coral reef, Filter feeders; Fluorescence microscopy; Caribbean

Funding: National Science Foundation Division of Ocean Sciences 1929293 This work was supported by the National Science Foundation Division of Ocean Sciences (No. 1929293). The funders had no role in study design, data collection and analysis, decision to publish, or preparation of the manuscript.

==============================
Microplastics (MP) are now considered ubiquitous across global aquatic environments. The ingestion of MP by fish and other marine vertebrates is well studied, but the ingestion of MP by marine invertebrates is not. Sponges (Phylum Porifera) are particularly understudied when it comes to MP ingestion, even though they are widely distributed across benthic habitats, can process large volumes of seawater, and can retain small particles within their water filtration systems. This study examines the presence of potential MP (PMP) in wild marine sponges and seawater collected in Bocas del Toro, Panamá. Subsurface seawater and tissue from six common Caribbean sponge species was collected in Saigon Bay, a heavily impacted, shallow-water coral reef. Seawater samples were filtered onto glass fiber filters to retain any PMP present and sponge tissue was digested with bleach, heated and filtered. Filters were examined using fluorescence microscopy to quantify PMP. An average of 107 ± 25 PMP L–1 was detected in seawater from Saigon Bay with particles ranging in size between 10 μm and ~3,000 μm. The number of PMP found in sponge tissue ranged between 6 ± 4 and 169 ± 71 PMP g–1 of dry tissue. Most particles found in sponge samples were very small (10–20 μm), but fibers greater than 5,000 μm were detected. Our results indicate that PMP exists within the tissues of the sponges we studied, but future studies should confirm the presence of MP in sponges using chemical analysis. Most importantly, the discrepancy between low levels of PMP in our sponge samples and high levels in the surrounding seawater highlights the potential for sponges to resist and/or egest MP. Finally, we provide a critical evaluation of our methods to improve their use in future MP work with benthic marine organisms.

Introduction

As humans continue to expand across the globe, our collective impact on the environment is amplified (Crutzen, 2002; Zalasiewicz et al., 2010; Lewis & Maslin, 2015). The detrimental effects of anthropogenic pollutants such as nutrients, chemicals, and sediment entering the environment via industrial effluents and agricultural runoff are well known, but the release of microplastics (MP) is of increasing concern (Browne, Galloway & Thompson, 2007; Thompson, 2015; Waller et al., 2017). Microplastics are defined as any plastic particle that is between 100 nm and 5 mm in size and include spheres, pellets, fibers and other small plastics commonly used in cosmetics, clothing, pharmaceuticals and industrial products (Zitko & Hanlon, 1991; Thompson et al., 2004; Betts, 2008; Arthur, Baker & Bamford, 2009; Koelmans, Besseling & Shim, 2015). They can be introduced to the environment via sewage, wastewater treatment effluents, industrial spills and runoff, and via the fragmentation of larger plastics (Browne et al., 2011; Cole et al., 2011; Conley et al., 2019). The progressive fragmentation of MP and their dynamic position in the water column due to wave action may impact planktonic, nektonic, and benthic organisms directly (Browne, Galloway & Thompson, 2007; Browne et al., 2008; Thompson et al., 2009; Wright, Thompson & Galloway, 2013). In addition, organisms encountering or consuming MP may be exposed to organic pollutants, heavy metals and pathogenic microbes bound to their surfaces (Mato et al., 2001; Hirai et al., 2011; Zettler, Mincer & Amaral-Zettler, 2013; Lamb et al., 2018; Rotjan et al., 2019; Dudek et al., 2020).

Much of the existing research on MP ingestion has revolved around vertebrates, with fish being the most studied group of aquatic organisms (de Sá et al., 2018). Studies that investigate MP ingestion by marine invertebrates are of mounting importance if we are to better understand the overall impact of MP in the marine world (Wright, Thompson & Galloway, 2013; de Sá et al., 2018). Marine ciliates (Christaki et al., 1998), calanoid copepods (Wilson, 1973), amphipods (Thompson et al., 2004), lugworms (Thompson et al., 2004), blue mussels (Ward & Targett, 1989; Ward, Levinton & Shumway, 2003; Browne et al., 2008; Ward & Kach, 2009), Pacific oysters (Sussarellu et al., 2015), sea cucumbers (Graham & Thompson, 2009), sea anemones (Morais et al., 2020), corals (Hall et al., 2015; Allen, Seymour & Rittschof, 2017; Rotjan et al., 2019), lobsters (Murray & Cowie, 2011) and the larvae of several invertebrate phyla (Hart, 1991) have been known to ingest MP in laboratory settings, and many ingest MP in situ. Several of these taxa exhibit some ability to select particles based on size or type, and some can defecate, regurgitate or otherwise egest the particles (Zebe & Schiedek, 1996; Wilson, 1973; Powell & Berry, 1990; Thompson et al., 2004; Graham & Thompson, 2009; Sussarellu et al., 2015; Hankins, Duffy & Drisco, 2018; Rotjan et al., 2019). Detrimental effects of MP ingestion by these animals include tissue inflammation, neurotoxicity, energy depletion, reduced skeletal growth rates, increased stress, and reduced immune function, feeding and reproduction (Besseling et al., 2013; von Moos, Burkhardt-Holm & Köhler, 2012; Avio et al., 2015; Cole et al., 2015; Sussarellu et al., 2015; Chapron et al., 2018; Hankins, Duffy & Drisco, 2018; Reichert et al., 2018; Tang et al., 2018; Rotjan et al., 2019). However, these impacts are highly variable across species, suggesting that some invertebrates may be more vulnerable to MP ingestion than others.

Sponges (Phylum Porifera) are particularly understudied in MP research, despite the fact that they are globally distributed across benthic ecosystems (van Soest et al., 2012; de Sá et al., 2018). Sponges are geographically and spatially cosmopolitan and occur in shallow-water reefs, deep-water habitats, hydrothermal vents, Polar regions, and in many freshwater systems (Hooper & van Soest, 2012). Sponges often exhibit high pumping rates (0.005–0.6 L of seawater s–1 L–1 of sponge tissue) and can therefore process large volumes of seawater through their canals and greater aquiferous systems (Reiswig, 1974; McMurray, Pawlik & Finelli, 2014; Pawlik, Loh & McMurray, 2018). In fact, sponge communities may overturn the water column (up to 30 m deep) every 1–56 days (Pile, Patterson & Witman, 1996; Savarese et al., 1997; McMurray, Pawlik & Finelli, 2014; Pawlik, Loh & McMurray, 2018). As marine sponges draw seawater through their systems of internal canals and chambers, they retain food particles including diatoms, cyanobacteria, viruses, flagellates, ciliates and yeast cells (Reiswig, 1971, 1974, 1975, 1990; Frost, 1978; Imsiecke, 1993; Pile, Patterson & Witman, 1996; Pile et al., 1997; Ribes, Coma & Gili, 1999; Kowalke, 2000; Hadas et al., 2006; Maldonado et al., 2010). These food particles are typically smaller than 70 μm in diameter (Ribes, Coma & Gili, 1999) because sponge ostia (exterior, incurrent openings) rarely exceed 60 μm and typically prohibit particles greater than 50 μm from entering the sponge (Reiswig, 1971; Simpson, 1984). The removal of these food particles from the water column by sponges plays an essential role in nutrient cycling on coral reefs (Lesser, 2006; van Soest et al., 2012; de Goeij et al., 2013; Pawlik, Burkepile & Thurber, 2016; de Goeij, Lesser & Pawlik, 2017). Importantly, as sponges increase their dominance on many coral reefs, their influence on overall reef function may become amplified (Zea, 1993; McMurray, Henkel & Pawlik, 2010; Colvard & Edmunds, 2011; Villamizar et al., 2013). Their widespread distribution, ability to retain small particles, and their prolific seawater filtering make sponges ideal candidates for evaluating MP abundance in marine systems.

Few studies have examined MP ingestion by sponges. One laboratory study exposed the temperate sponges Tethya bergquistae Hooper & Wiedenmayer (1994) and Crella incrustans Carter (1885) to 1 μm and 6 μm plastic beads and found no significant impact of the beads on sponge respiration or food particle retention (Baird, 2016). The study concluded that sponges may be resistant to MP exposure. Other laboratory studies have used plastic beads (0.1, 0.2, 0.5, 1.0, 4.0 and 5.7 μm in diameter) to study sponge physiology and have demonstrated the uptake of the beads in sponge tissues (Willenz & Van de Vyver, 1982; Turon, Galera & Uriz, 1997; Leys & Eerkes-Medrano, 2006). Recently, Girard et al. (2021) examined the presence, abundance and diversity of microparticulate pollutants in tropical sponges from North Sulawesi, Indonesia. They found that sponges do take up foreign particles, including MP such as polystyrene, and incorporate them into their skeletons and other internal tissues (Girard et al., 2021). The authors reported a maximum concentration of 612 foreign particles g–1 of dry sponge tissue, and concluded that sponges may act as bioindicators of marine microparticulate pollutants (Girard et al., 2021). Modica, Lanuza & García-Castrillo (2020) also recently found microfibers embedded on the surfaces of preserved museum sponge specimens representing 31 families. The authors predicted that the sponges, originally collected off the northern coast of Spain, were actively collecting fibers from the surrounding seawater and had been doing so for over 20 years (Modica, Lanuza & García-Castrillo, 2020).

Microplastics have been detected in the Caribbean and the sponges there are likely exposed to these contaminants. Bosker, Guaita & Behrens (2018) found an average of 261 MP kg–1 of sediment on four Lesser Antilles beaches while Acosta-Coley et al. (2019) found over 100 particles m–2 on some Colombian beaches. Garcés-Ordóñez et al. (2019) found up to 2,863 MP kg–1 of dry soil in polluted mangrove forests in Colombia while Rose & Webber (2019) found up to 0.00573 MP L–1 in surface seawater in the heavily polluted Kingston Harbor of Jamaica. However, surface measurements may seriously underestimate MP abundance (Gallo et al., 2018). For example, it is estimated that about one twelfth of the total number of MP present in the ocean ends up on the surface, with about the same fraction occurring in subsurface seawater and the rest occurring on the seafloor and on beaches (Andrady, 2011). Wright, Thompson & Galloway (2013) also noted that benthic suspension and deposit feeders may be exposed to biofouled and other high-density MP that sink to the benthos. Sponges are particularly abundant on Caribbean reefs with a high biomass, species diversity, and a percent cover that exceeds that of reef-building corals (Loh & Pawlik, 2014; Easson et al., 2015; de Bakker et al., 2017; Pawlik, Loh & McMurray, 2018). Many Caribbean sponges feed heterotrophically on particulate organic matter (POM) that is within the size range (<5 mm) of MP (McMurray et al., 2016; Rix et al., 2016). Together, these studies suggest that Caribbean sponge communities are likely exposed to MP pollution close to the benthos.

This study is the first to investigate the presence of potential MP (PMP) in Caribbean sponges from Bocas del Toro, Panamá and to report a subsurface PMP concentration in Caribbean seawater. We predicted that Saigon Bay, a heavily-impacted area in the Bocas del Toro archipelago, would be polluted with MP. We further predicted that marine sponges in the bay would be collecting these particles via filter feeding because sponges select food that is very small (<70 μm) and within the size range for particles considered to be MP (100 nm–5,000 μm). We used fluorescence microscopy to identify and quantify PMP. Additional identifying techniques such as FT-IR and Raman spectroscopy were not used because such technology was not available at the time of study. As such, our results represent only a survey of PMP (as per Covernton et al., 2019) in sponges from Bocas del Toro, and future studies should use chemical analysis to confirm the presence of MP in sponges. Still, we report the occurrence of PMP in six tropical sponge species and in seawater from Panamá and address the ecological implications of our findings.

Materials & methods

Study site and sample collection

Sponge and seawater samples were collected from Saigon Bay near Isla Colón, Bocas del Toro, Panamá (Fig. 1). Collection and export permits were approved by the Ministry of Environment of Panama (Research and Collections Permit No. SC/A-8-19 and Export Permit No. SEX/AO-1-2019). Saigon Bay sits immediately adjacent to houses, hotels and docks and is susceptible to anthropogenic pollution (Collin, 2005; Gochfeld, Schloder & Thacker, 2007; Easson et al., 2015; Fig. 1). The bay experiences a large degree of boat traffic, which may bring pollutants from other parts of the archipelago into the area. Bocas del Toro also has an underdeveloped waste disposal infrastructure (Aronson et al., 2004; Carruthers et al., 2005; Gochfeld, Schloder & Thacker, 2007; Easson et al., 2015). With heavy and frequent rains (3–5 m per year), much of this waste enters the surrounding waterways via runoff and some is even observed floating in the area (Carruthers et al., 2005; Collin, 2005; Kaufmann & Thompson, 2005; Gochfeld, Schloder & Thacker, 2007; Easson et al., 2015; B. Fallon, 2019, personal observations).

Figure 1 Map of Saigon Bay located off the coast of Isla Colón, the main island of the Bocas del Toro archipelago of Panamá.

The star indicates sample collection site. Note the high level of development on the northeastern border of Saigon Bay. Map data © 2020 CNES/Airbus © 2020 Europa Technologies © 2020 Google.

Sponge samples were collected on 21 June 2019 from ~6 to 8 m below the surface during an outbound tide so that any pollutants concentrated near the developed area were likely pulled through the bay (Fig. 1). A small (~5–8 cm; 0.08–1.0 g dry mass) section of sponge was removed by hand or by steel blade from three individuals (N = 3 replicates) of each of the six study species: Aplysina cauliformis Carter (1882), Amphimedon compressa Duchassaing de Fonbressin & Michelotti (1864), Callyspongia vaginalis Lamarck (1814), Ircinia campana Lamarck (1814), Mycale laevis Carter (1882) and Niphates erecta Duchassaing de Fonbressin & Michelotti (1864). These species were chosen as they represent some of the most dominant sponge species in the Caribbean (Loh & Pawlik, 2014) and include a diversity of growth forms. Each sponge section was wrapped in aluminum foil underwater and placed in a mesh bag for transport.

Four liters of seawater were collected at the same time and depth as the sponge samples. Four clean (washed with soap and water and rinsed three times with MilliQ®) 1 L glass jars were covered in foil and sealed with metal lids before descent. The jars were opened and filled at depth, and then re-covered with foil and sealed. This was replicated twice more on 1 July and 6 July 2019 at the same site during outbound tides. A total of three seawater samples (~4 L each) were obtained for this study. It should be noted that ~100–200 mL of seawater occasionally leaked from one of the glass jars. Thus, the total volume of seawater filtered for the seawater samples was between 3.6 and 4.0 L. Counts were normalized to seawater sample volume for quantification of PMP concentrations.

Sample processing

Seawater samples (N = 3) were processed separately on or close to their respective collection days (21 June, 1 July and 6 July 2019). Seawater (~4 L) was vacuum filtered onto a pre-combusted (450 °C for 4 h) 0.7 μm pore size (Whatman™ 1825-047 GF/F) glass microfiber filter. The four glass jars and sides of the filtration funnel were rinsed with analytical grade water (MilliQ®) and this excess water (~100 mL) was also filtered to maximize sample transfer. Seawater sample filters were then covered with another pre-combusted filter, wrapped in foil and stored at –20 °C until further analysis. Any PMP later found on the cover filters were added to the total number of PMP recorded for its corresponding seawater sample filter. A procedural blank was run in parallel to each of the three seawater samples, generating three MilliQ blanks (i.e., ~1 L of MilliQ was added to a clean beaker, filtered, and the filter was stored at –20 °C).

Each of the 18 individual sponge samples (three per species) was divided in approximately half using a steel utility blade: one half for preliminary analysis and methods development and the other half for final analysis. Each half was rinsed thoroughly with MilliQ (as we were only interested in PMP retained within the sponge body), weighed on a clean piece of foil, wrapped in foil and frozen at –20 °C until further analysis. The halves used for final PMP analysis were lyophilized and each sample was partitioned into three subsamples (~0.05–0.3 g dry) with a steel utility blade. Subsamples were used to minimize tissue digestion time and the dry mass of each subsample varied because sponge tissue density varied across species even though each original sponge sample was approximately equal in length (~2–4 cm). Final PMP counts were normalized to subsample dry mass during data analysis. Each subsample was cut into pieces with a utility blade and added to a clean 20 mL glass scintillation vial and covered with foil. In total, 54 subsamples were generated (6 species × 3 replicates × 3 subsamples). Approximately 5–10 mL of household bleach (Clorox®, 6% sodium hypochlorite) was added to each scintillation vial to digest the organic tissue. Bleach was used because it rapidly digests sponge tissue and because it causes minimal physical degradation of plastic particles (Hooper, 2003; Collard et al., 2015). The bleach we used was not pre-filtered to remove potential plastic contaminants before use because the high viscosity of bleach slows filtering time considerably, but we recommend filtering bleach before use for future studies (see “Evaluation of methods and considerations”). Still, we used procedural blanks (see following paragraph) to evaluate the degree of pre-existing experimental contamination in our samples. Vials with sponge tissue and bleach were heated to 60 °C for 2 h to expedite digestion (Conley et al., 2019; Payton, Beckingham & Dustan, 2020). If necessary, additional bleach was added to the vials to digest any remaining tissue.

After bleach digestion, each subsample (N = 54) was filtered onto a pre-combusted 0.7 μm pore size (Whatman™ 1825-047 GF/F) glass microfiber filter. Approximately 5–10 mL of MilliQ was added to the glass filtration funnel prior to the digested sponge subsample in order to minimize filtering time. After the sample was fully filtered, the sides of the funnel were rinsed with excess MilliQ to ensure maximum sample retention onto the filter. The filter was then removed and kept in a covered aluminum foil dish until further analysis. A total of six procedural blanks were run alongside the subsamples; approximately 10 mL of bleach was added to six clean scintillation vials, heated, and filtered.

Positive controls

Positive controls with known MP types were used to demonstrate plastic fluorescence behavior as well as the minimal effect of bleach and heat on that behavior. Microplastics were generated by cleaning common laboratory and consumer plastics (such as spray bottles, dish ware, monochromatic clothing, etc.) with 100% isopropyl alcohol and shaving particles (<5 mm) into clean 20 mL glass scintillation vials using a steel utility blade. Plastic type was identified by the recycling label or clothing tag on each unit of plastic. Ten plastic types were used including high density polyethylene (HDPE), low density polyethylene (LDPE), polyethylene (PE), polyethylene terephthalate (PETE), polypropylene (PP), polystyrene (PS), polyvinyl chloride (PVC), unknown plastic (recycling label #7) and the clothing fibers polyolefin and polyester (only clothes made with 100% polyolefin or polyester were sampled). Different colors of the same plastic type were collected when possible. Two scintillation vials were prepared for each plastic type: one for the addition of bleach and heat (N = 10 vials) and another for the addition of MilliQ without heat (N = 10 vials). Five to 10 mL of bleach was added to the first set of vials, which were then covered with foil and heated to 60 °C for 2 h. Additionally, 5–10 mL of MilliQ was added to the other set of vials, which were then covered with foil and not heated. All positive controls were filtered according to the sponge filtering procedure.

PMP visualization

All filters were analyzed for PMP presence using an E600 Nikon Eclipse fluorescence microscope fitted with a UV-1A fluorescence filter block (EX 360–370, DM 400, BA 400). Potential MP was distinguished from fluorescing background material (inorganic sand grains, proteinaceous spongin, invertebrate cuticle fragments, etc.) based on the brightness and color of fluorescence (Fig. 2; Fig. S1). Each filter was mounted onto a microscope slide with a few drops of MilliQ to secure the filter onto the slide. The filter was then surveyed at 100× total magnification in a sweeping motion vertically and laterally until the entire filter was visually surveyed, and the number of detected PMP was recorded. Because the seawater sample filters were so concentrated with PMP, each filter and its corresponding MilliQ blank was surveyed independently three times. The first survey resulted in counts that were consistently much greater than those resulting from the second and third surveys, while the latter two surveys resulted in consistently similar counts. Therefore, we concluded that the elevated counts from survey 1 were a result of human error, and the mean number of PMP for each seawater and MilliQ blank filter was determined using only counts from the second and third surveys. A 10 × 10 mm reticle net grid on one eyepiece in the microscope was used to measure the size of detected PMP. The size of nearly every PMP found in the sponge subsamples and corresponding blanks and the sizes of at least 15% of PMP found in the seawater samples and corresponding MilliQ blanks were recorded. The number and sizes of PMP in the positive controls were not recorded as they served only to demonstrate plastic fluorescence behavior and the effect of bleach and heat on that behavior. Only particles greater than or equal to 10 μm in maximum length were recorded for any filter. Particle sizes were categorized into nine groups based on maximum length: 10–20 μm, 21–50 μm, 51–100 μm, 101–300 μm, 301–500 μm, 501–1000 μm, 1001–3000 μm, 3001–5000 μm and >5000 μm. These size categories were chosen post hoc based on the sizes of the particles detected on the filters. They serve to demonstrate the frequency of different particle sizes in all samples and blanks, and to identify any differences in particle uptake or retention by sponges based on particle size. Particle sizes are reported in a stacked bar chart (Fig. 3) and do not reflect blank-corrected values (i.e., the proportion of particles within each size category represents the percent of total particles surveyed and may reflect the sizes of potential contaminants). The number of PMP on sponge and seawater sample filters was corrected based on the average number of PMP found on the corresponding blank filters. These corrections were not done on the basis of size (i.e., 10–20 μm particles in the blank were not subtracted from 10–20 μm particles in the sample) as we aimed only to evaluate general background contamination. Occasionally, this correction led to a negative value in some sponge subsamples, and in these cases, the PMP value for the sponge subsample was adjusted to zero.

Figure 2 Potential microplastics (PMP) on the filters of sponge samples, seawater samples and blanks.

The top two rows include samples from the six sponge species: (A) A. cauliformis. (B) A. compressa. (C) C. vaginalis. (D) I. campana. (E) M. laevis. (F) N. erecta. The bottom two rows include seawater samples (G–J) as well as one MilliQ blank from the seawater study (K) and one blank from the sponge study (L). Note the dulled, blue-green autofluorescence (indicated by white arrows) of spongin fragments, sand grains and two copepods in images A, D, G and I, respectively, as it compares with the bright, electric blue autofluorescence (indicated by red arrows) of PMP. Images were taken at 100× total magnification and the scale bar shown in “E” is applicable to all images.

Figure 3 The relative abundance (percent of total) of potential microplastic sizes detected in sponge and seawater samples and blanks.

Colors within the bars indicate the size of particles in micrometers (μm). Note the presence of large (3,001–5,000 μm) and very large (>5,000 μm) fibers only in some sponge samples.

Mitigating contamination

Since plastic is abundant in field and laboratory settings, several steps were taken to minimize sample contamination. Nitrile gloves and 100% cotton cloths and lab coats were used at all times during sample processing. However, it is possible that cotton fibers from these materials were counted in blanks and samples because cotton (cellulose) may autofluoresce under UV light (Malinowska et al., 2015). The particular fluorescence behavior of cotton cellulose under our microscopy conditions was not tested in this study. Glassware was used in place of plasticware and all glassware and samples were covered with foil when not in use. Glassware and metal utensils were cleaned with soap and water and rinsed three times with MilliQ before use. Lastly, dry sponge samples were cut into scintillation vials under a laminar flow hood to reduce airborne contamination. Though these steps were taken to minimize contamination, we recognize that false positives are still possible.

Data analysis

Potential microplastic (PMP) concentrations are reported as number of PMP per liter (PMP L–1) for seawater samples and number of PMP per gram of dry tissue (PMP g–1) for sponge samples. The blank-corrected number of PMP g–1 was determined for each of the 54 subsamples representing the six species. These 54 values were then grouped by replicate to produce a mean number of PMP g–1 for each replicate. These true replicate values (N = 3) were then grouped by species to produce a mean number of PMP g–1 for each species. A one-way ANOVA test followed by a Tukey’s HSD pairwise multiple comparisons test was used in R (R Development Core Team, 2019) to determine any significant differences in mean PMP concentrations between the six sponge species.

Results

Plastic fluorescence behavior and positive controls

Plastic particles fluoresced electric blue when exposed to UV light (except for red PP, which fluoresced pink) and often fluoresced much brighter when compared to other materials (sand grains, spongin, chitin, etc.), which had a blue-green and dulled fluorescence (Fig. 2; Fig. S1). Some plastic types (e.g., HDPE, PVC) showed weak to no fluorescence, while others (e.g., unknown plastic with recycling label #7, PETE, Polyester clothing fibers, PP) showed intermediate to strong fluorescence (Fig. S1). Lightly colored plastics (e.g., clear, white, yellow, light blue, red) fluoresced more often and stronger than darkly colored plastics (e.g., black, brown, gray, green, dark blue), though some light plastics did not fluoresce at all (Fig. S1). Some plastics (i.e., unknown plastic with recycling label #7) in the control MilliQ set showed small flecks of fluorescent material even if the plastic itself did not fluoresce (Figs. S1G and S1H). Exposure to bleach and heat did not impact plastic fluorescence behavior (Fig. S1).

Seawater

An average of 107 ± 25 PMP L–1 was found in seawater samples collected from Saigon Bay. The PMP detected in seawater varied in size, though very small (10–20 μm) particles comprised ~25% of the total number of particles detected (Fig. 3). The corresponding MilliQ blanks had proportionately fewer very small particles (~16%), with ~27% of all particles in the blanks being 101–300 μm in maximum length (Fig. 3). Although one large fiber (3,001–5,000 μm) was found in the seawater samples, no large fibers were found in the blanks, and no very large fibers, or those that were >5,000 μm and technically outside the range of MP, were found in the seawater samples or MilliQ blanks. The number of particles detected in the MilliQ blanks (no more than eight particles per filter) never exceeded 10% of those found in the samples, indicating that there was minimal contamination during sample processing (Gago et al., 2016).

Sponge

The number of PMP g–1 of dry tissue varied across the six sponge species (one-way ANOVA, df = 5, F = 5.358, p = 0.0081; Fig. 4, Table 1). Callyspongia vaginalis, A. cauliformis, N. erecta and I. campana showed the highest concentrations of particles (mean ± SE 169 ± 71, 113 ± 23, 75 ± 38 and 71 ± 20 PMP g–1, respectively; Fig. 4), while A. compressa and M. laevis showed lower concentrations (14 ± 2 and 6 ± 4 PMP g–1, respectively; Fig. 4). However, there were few significant pairwise differences in mean PMP concentration between the six species (Fig. 4; Table 1). The number of particles in the procedural blanks was sometimes greater than that in the sponge subsamples themselves; seventeen of the 54 subsamples had a negative net number of particles. As such, PMP counts in the sponge blanks as a percentage of counts in the sponge subsamples sometimes exceeded 100%. This finding is concerning because a blank percent of 10% has previously been used as a threshold to signify that sample counts are significantly greater than blank counts (Gago et al., 2016). A relatively high level of background PMP in our subsamples was likely the result of using non-filtered bleach and/or the use of very small amounts of tissue (~0.05–0.3 g) for each subsample (see “Evaluation of methods and considerations”).

Figure 4 Number of potential microplastics (PMP) g–1 of dry sponge tissue.

Box plots (N = 3 replicates per species) are median inclusive and the “x” in each plot represents the mean. The mean number (±SE) of PMP g–1 of dry tissue for each species is listed below the sponge name. For reference, the mean number of PMP found in the seawater samples was 107 ± 25 particles L–1. Letters above each plot indicate significant pairwise difference (Tukey’s test, p < 0.05).

Table 1 Results from a Tukey’s HSD pairwise multiple comparisons test of mean potential microplastic (PMP) abundance across sponge species in R (R Development Core Team, 2019).

The six sponge species are Aplysina cauliformis, Amphimedon compressa, Callyspongia vaginalis, Ircinia campana, Mycale laevis and Niphates erecta and significant differences (p < 0.05) are boldfaced.

Pairwise comparison	Adjusted p-value	
A. cauliformis - A. compressa	0.1269	
A. cauliformis - C. vaginalis	0.9701	
A. cauliformis - I. campana	0.9291	
A. cauliformis - M. laevis	0.0360	
A. cauliformis - N. erecta	0.8863	
A. compressa - C. vaginalis	0.0365	
A. compressa - I. campana	0.4709	
A. compressa - M. laevis	0.9687	
A. compressa - N. erecta	0.5416	
C. vaginalis - I. campana	0.5608	
C. vaginalis - M. laevis	0.0101	
C. vaginalis - N. erecta	0.4893	
I. campana - M. laevis	0.1661	
I. campana - N. erecta	0.9999	
M. laevis - N. erecta	0.2016	

Potential MP found in the sponge samples and blanks were mostly very small (10–20 μm) or small (21–50 μm), and together comprised over half of the total number of PMP detected in all sponge samples (52%) and blanks (66%) (Fig. 3). Larger particles (51–1,000 μm) comprised 35% of the total number of particles in all sponge samples, whereas they made up 25% of those found in the blanks (Fig. 3). Medium (1,001–3,000 μm) fibers comprised 10% and 9% of particles in the sponges and blanks, respectively. Large (3,001–5,000 μm) and very large fibers (>5,000 μm; technically outside the range of MP) were occasionally (3% of all particles) found in the sponge samples (to the exclusion of A. compressa), but were never detected in the blanks (Fig. 3).

Discussion

Potential microplastic in seawater

An average concentration of 107 ± 25 PMP L–1 of seawater in Saigon Bay is striking. Few studies have investigated MP concentrations in Caribbean seawater but reports of surface concentrations have not exceeded 0.00573 MP L–1 (Law et al., 2010; Rose & Webber, 2019). These previous studies targeted larger particles that were >335 μm and collected using plankton tows, while we targeted smaller particles that were >10 μm and collected as bulk samples close to the benthos. Surface MP concentrations in the world’s coastal waters and oceans are also reported as lower than ours, though these studies again targeted larger size fractions. Colton, Knapp & Burns (1974) reported a concentration of 0.000067 MP L–1 (>947 μm, plankton tows) in the open northwest Atlantic Ocean while Doyle et al. (2011) reported a maximum of 0.00019 MP L–1 (>505 μm, plankton tows) in the coastal northeast Pacific Ocean. Aliabad, Nassiri & Kor (2019) reported a maximum of 0.00114 MP L–1 (>333 μm, plankton tows) in the Gulf of Oman while Payton, Beckingham & Dustan (2020) reported a maximum of 0.6 MP L–1 (>43 μm, grab samples) in the estuarine Cooper River of South Carolina.

Recent findings suggest that previous studies significantly underestimate MP concentrations in seawater because plankton tow nets (300–1,000 μm mesh) are commonly used when sampling the upper 1 m of the water column (Covernton et al., 2019). Kang et al. (2015) and Barrows et al. (2017) concluded that these tows allow smaller MP (<300 μm) and fibers (due to their small width) to pass through holes in the nets, and that these studies may be underestimating seawater MP concentrations by orders of magnitude. Covernton et al. (2019) compared the suitability of in situ sieve versus bulk sample methods to measure MP abundance in seawater. They found that bulk seawater samples collected in 1 L glass jars and filtered directly onto 8 μm pore size filters resulted in PMP concentrations that were on average 8.5 times higher than samples that were collected in 10 L buckets, sieved using a 63 μm mesh in the field, and then filtered (Covernton et al., 2019). They concluded that studies using plankton nets may underestimate MP concentrations by up to four orders of magnitude compared to studies that target smaller (<100 μm) plastics (Covernton et al., 2019). The authors emphasized the necessity of using bulk seawater samples and sensitive filtration methods (ability to collect plastics down to 10 μm) when assessing the exposure of marine organisms to MP pollution (Covernton et al., 2019).

An average seawater concentration of 107 PMP L–1 in our study compares better with MP studies that used bulk samples and sensitive filtration methods. Covernton et al. (2019) reported 5.28 PMP L–1 in filtered (8 μm) surface seawater samples from coastal British Columbia, Canada. Jiang et al. (2020) pumped surface seawater through 50 μm net and reported 6.5 MP L–1 in the South Yellow Sea. Norén & Naustvoll (2010) pumped surface seawater through a 10 μm filter and reported 102 MP L–1 in Swedish coastal waters. Although our subsurface value of 107 PMP L–1 corroborates Norén & Naustvoll (2010), comparisons with previous studies are difficult. Our result reflects the presence of PMP in seawater from Saigon Bay, and future studies are needed to verify the presence of MP in this area using chemical analysis. Additionally, we sampled subsurface seawater and used different methods (grab samples filtered directly onto a 0.7 μm filter) compared to these previous studies, so it is uncertain whether our seawater value is relatively high compared to reported values. As such, subsurface seawater must be evaluated at additional sites and across time in the Bocas del Toro archipelago to confirm an elevated level of plastic contaminants in this area compared to previous reports. We recommend the continued use of bulk samples and sensitive filtration, coupled with higher resolution techniques, to more accurately identify and quantify MP particles in seawater.

Potential microplastic in sponges

This is the first study to evaluate the presence of MP in wild Caribbean sponges. Although we used only fluorescence microscopy to quantify PMP within our sponge samples, our blank-corrected values offer important insight into the potential for wild sponges to ingest plastic particles. The abundance of PMP detected in our sponge samples (6–169 PMP g–1) compares well with or was lower than that detected in a recent study. Girard et al. (2021) examined microparticulate pollutants (including natural particles like minerals and shell fragments, as well as anthropogenic particles like particulate cotton and polystyrene) in tropical sponges from Indonesia. Like us, the authors used bleach-digested dried sponge subsamples (0.0022–0.011 g dry) and vacuum-filtered them onto 1 μm pore size membranes (Girard et al., 2021). Using Raman spectroscopy, they detected 91–612 foreign particles g–1 (combination of natural and man-made) ranging in size from 5 to 200 μm in the sponge subsamples (Girard et al., 2021). Of note was one sponge, Ircinia sp., which was found to have 159 polystyrene MP g–1 of dry tissue (Girard et al., 2021). As such, our results (6–169 PMP g–1) align well with those of Girard et al. (2021), especially considering that the PMP detected in our samples would only be a fraction of the total particles detected in their samples. However, this comparison is limited as our results indicate the presence of PMP in our sponge samples, whereas Girard et al. (2021) confirmed the presence of MP in their samples using Raman spectroscopy.

Much of the PMP (42%) found in our sponge samples was very small (10–20 μm), while PMP of the same size range made up only about one quarter of those found in seawater. This suggests that the sponges we studied may demonstrate some selectivity in PMP ingestion, preferring very small particles. This finding is not surprising considering that sponges typically feed on microorganisms smaller than 70 μm (Ribes, Coma & Gili, 1999). Moreover, laboratory studies have demonstrated the retention of microbeads (<5.7 μm) in sponge tissues, which supports the idea that sponges prefer very small particles (Schmidt, 1970; Willenz & Van de Vyver, 1982; Imsiecke, 1993). Still, a significant percent (~35%) of all particles (excluding fibers) found in our sponge samples were between 51 and 1,000 μm in maximum length. Because previous studies have demonstrated that sponges tend to ingest particles smaller than 50 μm (Reiswig, 1971; Simpson, 1984), it is difficult to perceive at present the mechanism by which such large particles may be entering sponge bodies. However, Cerrano et al. (2002) found that exopinacocytes on the surface of some Indonesian sponges can endocytose particles as large as 2 mm into the sponge body. Other studies have also documented endocytosis as an important ingestion process in sponges (Willenz & Van de Vyver, 1982; Hammel & Nickel, 2014, Girard et al., 2021), and new studies should continue to investigate potential uptake processes and confirm the presence of such large particles in sponges.

A total of 20 large fibers (3,001 μm to >5,000) were detected in the sponge samples but they were absent in the blanks and present only once in the seawater samples. This finding suggests that the sponges we sampled may concentrate synthetic fibers from seawater as they filter feed. It is plausible that the fibers we detected were endocytosed from the ectosome by action of exopinacocytes (Cerrano et al., 2002), or that they passed through the sponge ostia and became stuck in the sponges’ internal canals because fiber width, regardless of maximum length, never exceeded 10 μm. In addition, we rinsed the outside of our samples prior to analysis in an attempt to isolate particles retained only within the sponge aquiferous system. Still, it is possible that the fibers we detected were embedded on the surface of the sponges (Modica, Lanuza & García-Castrillo, 2020). It should also be noted that rinsing the sponge samples prior to analysis may have removed particles from sponge oscula that were exiting the sponge body at the time of collection, and future studies should consider this when evaluating particle egestion.

The location of PMP within the bodies of the six sponge species we studied is unknown, but recent studies have highlighted the presence of microparticulate pollutants in the ectosome (outer layer of the sponge body), inner mesohyl, and around the choanocyte chambers of northern Atlantic and western Pacific sponges (Modica, Lanuza & García-Castrillo, 2020; Girard et al., 2021). The latter study predicted that some particles were captured on the sponge surface by exopinacocytes and were subsequently drawn into the body, while other particles were drawn passively into the aquiferous system via ostia and were later phagocytized by choanocytes (Girard et al., 2021). The authors also suggested that non-spiculate sponges tended to incorporate larger (>50 μm) particles into their skeletons whereas spiculate sponges tended to incorporate smaller (<50 μm) particles into their ectosome (Girard et al., 2021). We did not perform histological experiments in our study and so cannot report the location of PMP within the tissues of our study species. However, because we examined both non-spiculate (A. cauliformis and I. campana) and spiculate (A. compressa, C. vaginalis, M. laevis and N. erecta) sponges, it is possible that these species may be incorporating MP into their tissues in ways suggested by Girard et al. (2021). Furthermore, the calcareous sponge Sycon coactum Urban (1906) has been shown to egest microbeads (up to 1.0 μm) by action of choanocytes, which can engulf the beads and carry them into excurrent chambers (Leys & Eerkes-Medrano, 2006). The ability of additional sponge species to egest other particles like algal remnants, bacterial cells and sediment is also well documented (Francis & Poirrier, 1986; Imsiecke, 1993; Yahel et al., 2007, McGrath et al., 2017), and demonstrates the potential for sponges to egest MP. However, the ability of our specific sponge species to egest particles is unknown. Future work should use histological methods to better understand how MP enter the sponge body, where they are being retained, and whether more species can egest MP.

Variation in PMP concentration and composition across sponge species may relate to differences in sponge morphology, physiology, and/or the availability of MP in seawater. Sponge traits such as ostia/oscula diameter, tissue density, pumping rate, aquiferous system complexity, and/or microbial abundance may impact MP uptake and retention because these traits impact the volume and residence time of seawater processed by sponges, as well as the size of particles allowed into the sponge body (Reiswig, 1974; Weisz, Lindquist & Martens, 2008; Easson et al., 2015). Interestingly, Girard et al. (2021) observed that particle incorporation by sponges was independent of particle material. In other words, the authors suggested that the sponges would take up particles based on what was available in the surrounding seawater, and that any differences in the composition of incorporated particles between species depended only on particle spatial variation (Girard et al., 2021). Similarly, Modica, Lanuza & García-Castrillo (2020) found that fiber abundance in sponge ectosomes was independent of sponge species, habitat type and depth, and that fibers were likely ubiquitous in the surrounding seawater. Like these previous studies, our results indicate that particle uptake (amount and size) is independent of sponge species. There were few significant differences in mean PMP concentration between our sponge species (Fig. 4; Table 1), suggesting that they showed no differences in their susceptibility to PMP ingestion. Additionally, the size fractions of particles detected in our samples were approximately equal across all species (Fig. 3), indicating that one species did not prefer particles of a particular size compared to another species. Thus, our findings support the hypothesis that particle uptake and retention (amount and type) is independent of sponge species (Modica, Lanuza & García-Castrillo, 2020). Furthermore, because there is little detailed data describing the morphological and physiological features of the specific sponge species we studied (e.g., ostia/oscula size, tissue density, pumping rate, etc.), and because we did not measure these parameters in our study, we cannot yet draw correlations between these characteristics and any differences found in PMP loads. Future studies should aim to identify any such relationships across ours and additional sponge species. Also, studies should compare MP loads and types in samples collected from areas that are differentially impacted by MP pollution to confirm whether loads in sponges depend on particle spatial variation.

Ecological implications

A relatively high concentration of PMP in seawater from Bocas del Toro indicates that marine life in the area may be exposed to MP. The archipelago is home to numerous marine invertebrate species and many commercial and non-commercial fishes (Collin, 2005; Seemann et al., 2014). High PMP concentrations in the archipelago’s coastal waters suggests that these local species may be susceptible to MP ingestion. This risk will likely be exacerbated as tourism and residency rates continue to climb (Easson et al., 2015; The World Bank, 2018; Dorsett & Rubio-Cisneros, 2019).

Scaling our data to appreciable values helps to illuminate the story of PMP in sponges collected from a heavily impacted reef in Bocas del Toro. An average concentration of 87 PMP g–1 across all sponge species in this study equates to >8,000 PMP particles in a sponge that weighs 100 g (dry), or a sponge that is approximately 1.5 L (McMurray, Blum & Pawlik, 2008; Girard et al., 2021). This number agrees well with that reported by Girard et al. (2021) who predicted that at least 10,000 microparticulates (sum of MP, minerals, etc.) per sponge may exist in some demosponges (100 g dry) from Indonesia. Furthermore, using known pumping rates (~0.09–0.48 L s–1 L–1) and tissue densities (~89–155 g L–1) for sponges that are congeneric with our species (Weisz, Lindquist & Martens, 2008; Fiore, Freeman & Kujawinski, 2017; Pawlik, Loh & McMurray, 2018), and an ambient seawater concentration of 107 PMP L–1 (our study), we would predict that a 100 g sponge could be passing between 25,000 and 174,000 particles through its body every hour. These values are far greater than the 8,000 PMP we predict to be present in a sponge at any given moment in time. This finding supports the hypothesis that the sponges we studied have some capacity to resist MP ingestion and/or that they have some ability to egest the particles.

Interestingly, despite the presence of PMP in every species, the sponges from which samples were taken appeared to be healthy and functional, as they had open ostia, showed no evidence of necrosis, and were large individuals. Based on this gross examination, we did not detect an external effect of PMP ingestion in sponges from Saigon Bay. From laboratory experiments, Baird (2016) also reported an absence of effect with MP exposure having little impact on temperate sponge respiration. In addition, relatively low concentrations of PMP in sponge tissue despite there being ~107 PMP L–1 in seawater in Saigon Bay support the idea that tropical sponges have some capacity to resist and/or egest MP. As selective filter feeders, perhaps sponges can adjust their pumping rates in response to pulses of MP, as sometimes occurs with increased sediment load (Gerrodette & Flechsig, 1979; Maldonado et al., 2010; McMurray et al., 2016). Girard et al. (2021) suggested that sponges may act as bioindicators of general microparticulate pollutants, but our results indicate that marine sponges may be resistant specifically to MP exposure and therefore may not be the best indicators of MP pollution. Increased spatial and temporal sampling is needed to test the potential for sponges to act as bioindicators of MP in aquatic environments.

Evaluation of methods and considerations

We acknowledge that the methods used in this study have some limitations. Only fluorescence microscopy was used to identify and quantify suspected MP. The lack of secondary verification, such as by Raman or FT-IR spectroscopy, requires us to refer to the particles detected as potential microplastics (PMP) as per Covernton et al. (2019). The sole use of fluorescence microscopy to identify PMP in our samples raises several concerns. Firstly, the lack of additional verification methods means that the number of particles detected in our samples may be positively skewed owing to false positives. However, Payton, Beckingham & Dustan (2020) noted that fewer MP in seawater samples were detected using fluorescence microscopy than when using brightfield microscopy alone, indicating the potential also for some negative bias in our results because we did not use brightfield microscopy. In our positive controls, we confirmed that not all plastic types fluoresce under our microscopy conditions, and that there is variation in fluorescence strength and color between plastic types. Since we only counted particles that fluoresced strongly with an electric blue color (i.e., the fluorescence behavior of white and clear fragments of PETE and PP), our results may reflect the presence of only particular plastic types and therefore underestimate the true number of MP present in the sponge and seawater samples. It is also possible that we counted non-plastic particles that also exhibited this fluorescence behavior. We strongly recommend the use of additional verification methods in future studies to confirm the presence of MP in sponges and other organisms. Recently, fluorescence lifetime imaging microscopy (FLIM) has been shown to confidently identify MP in biological samples (Monteleone et al., 2021), and warrants consideration for future use in MP studies.

We also recognize that an appreciable number of particles were found in the blanks for the sponge study, sometimes amounting to more than were found in the sponge subsamples themselves. As such, we report the presence of PMP in sponges from Bocas del Toro with caution. While the counts in the MilliQ blanks as a percentage of counts in the seawater samples was low (<10%), MilliQ blanks were not digested with bleach. This suggests that MP present in commercial bleach may have created a higher PMP background level in our sponge subsamples. This background contamination would likely be reduced by pre-filtering the bleach solution, and furthermore it’s contribution to sample counts would be diminished if larger dry tissue subsamples (>0.3 g dry) were analyzed. As such, our study highlights the necessity in filtering all solutions before use in MP studies. However, even if some of the PMP in the sponge samples are artifacts of PMP in bleach, it is most striking how few particles were detected in the sponge samples compared to the high load that was detected in the surrounding seawater, especially considering that sponges process such large volumes of water as they filter feed.

Finally, the digestion method used in this study offers an efficient and cost-effective way to survey marine sponges for the presence of PMP. The use of bleach with heat to digest organic material showed no impact on the physical integrity or fluorescence behavior of plastic particles. This digestion method also agrees with Collard et al. (2015) and other recent studies (J. Lynch, 2019, personal communication) and we recommend its use in future MP studies, with the addition of a pre-filtering step.

Conclusions

This study surveys the occurrence of PMP in wild sponges and in subsurface seawater from Bocas del Toro, Panamá. Digestion of dry tissue using household bleach is a time- and cost-effective method for evaluating MP presence because it showed no effect on plastic physical integrity or fluorescence behavior. We recommend this technique with some additional solution preparation as potential background contamination in our sponge samples highlights the need to filter bleach before use in MP work. Additionally, the use of only fluorescence microscopy to identify and quantify PMP in our samples limits our ability to compare our results with those of previous MP studies, and we strongly encourage the use of additional chemical analysis in future work to confirm the identity of MP. Still, a PMP concentration of ~107 PMP L–1 in subsurface seawater from Saigon Bay reported in this study compares well with or is greater than previous reports of MP that used bulk samples and sensitive filtration techniques (down to 10 μm). As such, we recommend the continued use of these methods along with the use of subsurface samples when evaluating the exposure of benthic filter-feeding organisms to MP. Our results further indicate that sponges ingest PMP, and that the sponges we studied may preferentially collect fibers and very small (10–20 μm) particles. However, the occurrence of PMP (6–169 PMP g–1) in tissue samples taken from seemingly healthy sponges was relatively low considering the high level of PMP detected in seawater. This finding suggests that the sponges we studied may be somewhat resistant to PMP ingestion and/or have some capacity to egest the particles. Lastly, the presence of PMP in sponges and seawater from Saigon Bay suggests that marine life may be exposed to MP in Bocas del Toro. This exposure is only expected to increase with a growth in population and tourism. Our study highlights the lack of MP research in the Caribbean, and future work should aim to further evaluate the presence and impact of MP in this beloved and highly-frequented region.

Supplemental Information

Supplemental Information 1 Fluorescence behavior of known plastic types.

(A, a) white HDPE. (B, b) brown HDPE. (C, c) yellow PP. (D, d) white PP. (E, e) red PP. (F, f) white PVC. (G, g) white unknown (recycling label #7). (H, h) brown unknown. (I, i) white PETE. (J, j) polyester. Capital letters indicate plastics that were added to MilliQ and were not heated, whereas lowercase letters indicate plastics that were bleached and heated. Note the minimum effect of bleach and heat on physical integrity and fluorescence behavior. Images were taken at 100× total magnification and the scale bar shown in “e” is applicable to all images.

Click here for additional data file.

Supplemental Information 2 Abundance of potential microplastic (PMP) in seawater and sponges from Bocas del Toro, Panamá.

Click here for additional data file.

Supplemental Information 3 Data analysis for determining the abundance of potential microplastics (PMP) in seawater and sponges and calculations to support scaling arguments.

Click here for additional data file.

Supplemental Information 4 Code and results for data analysis in R.

Click here for additional data file.

Supplemental Information 5 Results of data analysis in R.

Click here for additional data file.

We would like to thank the Smithsonian Tropical Research Institute for lab space, field supplies and boat access. We also thank the Hollings Marine Laboratory, College of Charleston, Grice Marine Laboratory, M. Janech, A. Bland, P. Lee and N. Schanke for technical support, lab space and supplies. Thanks to C. Easson, C. Fiore, D. Gonzalez, S. Czwalina, A. Stephens and J. Thurnham for their assistance with sample collection. We also thank S. Czwalina for assistance with sample processing and A. Parry for help with data analysis. Thanks to the editor J. Pawlik, to the reviewer C. Motti, and to two anonymous reviewers for their constructive comments. Finally, this research would not have been possible without the advice and guidance of B. Beckingham, J. Lynch, L. Jonas, K. Dudek, C. Fiore, C. Easson, G. Lôbo-Hajdu, M. Janech, R. Thacker and P. Dustan.

Additional Information and Declarations

Competing Interests

Author Contributions

Field Study Permissions

Data Availability

The authors declare that they have no competing interests.

Bailey R. Fallon conceived and designed the experiments, performed the experiments, analyzed the data, prepared figures and/or tables, authored or reviewed drafts of the paper, and approved the final draft.

Christopher J. Freeman conceived and designed the experiments, analyzed the data, authored or reviewed drafts of the paper, and approved the final draft.

The following information was supplied relating to field study approvals (i.e., approving body and any reference numbers):

Collection and export permits were approved by the Ministry of Environment of Panama (Research and Collections Permit No. SC/A-8-19 and Export Permit No. SEX/AO-1-2019).

The following information was supplied regarding data availability:

The data are available in the Supplemental Files.

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
