# Peer review of "Plastics in Porifera: The occurrence of potential microplastics in marine sponges and seawater from Bocas del Toro, Panamá"

_PeerJ, doi:10.7717/peerj.11638_

## Round 0.1 · original submission · Major Revisions

Dear Chris and Bailey,

I now have 3 reviews from researchers with expertise both in sponges and plastics, and the majority decision is that this contribution will require major revisions prior to potential publication. While the reviewers acknowledge the authors' transparency regarding the limitations of the methods used in this study, they also see value in publishing this, provided the authors are clear, and the conclusions are circumspect. I hope you agree that the reviewers have provided thoughtful and detailed reviews, and I trust that the revision will reflect their comments.

Reviewer 1 ·

Basic reporting

Overall the writing was good, with pertinent background information, supporting citations, and a central question.

There were a few odd/confusing statements, for example:

"Like sponges, MP is also likely common in the Caribbean" (line 138-139).

The methods section says that "one procedural blank was run along with each of the three water samples" (line 201-202) which I initially interpreted as one blank being run per three water samples, rather than one blank per sample.

In results, would have liked you to report number of PMPs in blanks (for example in line 320)

.

Experimental design

In general, the sample size of 3 for water samples and each sponge species is small. .
Additionally, six procedural blanks for 54 subsamples is not particularly strong. The counts for blanks and sponge samples varied per subsample set (as seen in the raw data, in which there were very few particles counted in samples/blanks in the first subset and greater numbers counted in the second subset.

Why weren't the dry masses of sponge samples more standardized?

The volume used for the water procedural blank samples (1L) would be stronger if it had been the same volume as the water samples (4L).

The microscopy method could have been more clear- how were filters mounted and and how were the measurements conducted?

Validity of the findings

You acknowledge the limitations of the methods in terms of plastic identification and the designation of fluorescent particles as "potential" microplastics. Were any other identification methods attempted (looking for striations, uniform diameter in fibers, or other qualitative properties)? In the discussion, it should be acknowledged that discrepancies between these results and other reported plastic concentrations in seawater may be due to the identification of any fluorescent particle as PMP and not only the sensitive filtration method (line 354-355/line 391). The positive control tests of known plastic types is largely qualitative- it doesn't seem to aid much in the plastic identification in actual samples. Were other non-plastic items tested to determine how they fluoresced?

The introduction states that the size of the ostia generally prevent particles larger than 50um from entering the sponge, yet you found many >50um particles in the sponge. You touch on this in the discussion, how fiber width didn’t exceed 10um (line 421-422), but the mechanism for ingestion of this type of particle is difficult to conceive.

In looking at the metadata file, the blank % in samples as a whole was high- about half of the subsamples had >75% of particles potentially coming from the methodology itself. Is there a significant difference between number of particles in sponge samples vs. the associated blanks?

The metadata file also shows that there were different viewing trials for the water samples that were not described in the methods, with the average taken from viewing trials 2 and 3 (and not 1)? This is unclear.

Additional comments

The study addresses a gap in marine plastics research. The inability to positively identify plastics, the small sample sizes and number of blanks run doesn't provide the strongest evidence, but you are very transparent about the study's limitations and cite other published works using similar techniques.

Reviewer 2 ·

Basic reporting

I commend the authors on submitting a well-written and composed manuscript. The ample citations and clear writing make assessing the content easy and will increase reader comprehension. I’m not sure what the recommendations are for PeerJ, but there are some places where there are a large number of citations listed that may not be necessary, however this is not necessary to change, just something I took notice of.

Experimental design

While the overall design is acceptable, I do have a small concern that I'd like for you to address in the text. The question/hypothesis (what is the prevalence of potential microplastics in sponges and seawater in the Caribbean?) is generalizing across the entire Caribbean sponge population and the nearshore coastal seawater of the Caribbean Sea, yet the samples come from just one location in one area of the Caribbean and may not represent the Caribbean population as a whole (sample does not properly estimate the population parameter of interest as currently phrased). I would caution against extrapolating your conclusions and statements to the entire Caribbean. This location is a sheltered archipelago that is largely outside of the currents and circulation patterns seen in the Caribbean as a whole, and I'd argue that this location should not be used to speculate what the entire Caribbean is experiencing. It's slightly misleading to draw conclusions for the entire Caribbean from a single site. Especially when the standard errors suggest that sample size could be larger to better estimate the population parameter.

Validity of the findings

No comment (but see comment regarding extrapolation to the wider Caribbean above)

Additional comments

Very well-written manuscript, thank you! One suggestion, I'd consider adding the sample size to the figure captions where appropriate (ex. Figure 4) so that the reader knows how many samples are being summarized. Will the data be provided for public use? If yes, consider formatting in long form so that it's easily imported into statistical software.

·

Basic reporting

Overall, the manuscript is well written and the English adequate. However, there are several areas that need to be addressed – detailed below.

A) Units and terminology:
1) Units throughout are inconsistent. Sometimes you write ‘per liter’ other times ‘L-1’ and other times ‘/L’. I suggest that you use ‘L-1’ and ‘g-1 dry tissue’. Where this occurs, I have highlighted on the marked up PDF. Please change MP/kg to MP kg-1 throughout.
2) Only use full unit description when at the beginning of a sentence. This is done correctly on Line 184, but incorrectly on line 173 – should read ‘from ~6-8 m below…”..
3) Please change hour to hr.
4) Please check all ml are mL to be consistent with L used elsewhere.
5) Terminology switches between water and seawater. Please use correct prefix when when referring to fresh water or shallow/deep water, waste water etc, and use seawater when referring to the samples collected from the ocean (either in your study or others you are citing). For example,
a. Line 39 should read “Seawater samples were filtered….”
b. Line 131 should read “from the surrounding seawater….”
c. Line 143 should read “in surface seawater…”
d. Line 184 should read “Four liters of seawater were collected…”
e. And so on….
This is important as on line 320 “in the water samples or water blanks” it is unclear whether you are referring to a seawater blank or a Milli-Q blank – I presume it should read “in the seawater samples or MilliQ blanks” – please clarify.
Please check the main text and all figures and tables regarding the above – please make everything consistent.

B) Literature references are adequate. However, Please check volume and page numbers of the following references
Avio
Crutzen
Girard
Law

Also, there is a space between Dudek and Easson – might be an issue with citation software, or simply an extra return – please remove.

C) The structure and layout is adequate. Raw data is adequate.

D) Tables and Figures:

Table 1 and Figure 4 should be combined. They show the same data. I suggest keeping Figure 4, but elevating to position of Figure 2. Include the numbers from the table on the graph (or under the sponge name). For the seawater sample add this data to the legend. For example: “For reference, the mean PMP L-1 in seawater samples is 107(+/- 25).”

Also, regarding the dots indicating outliers, these are difficult to see, I suggest changing them to a different symbol, i.e., diamond – as there are no diamond shapes associated with the sponge images.

Please provide full genus name for all sponges. Change water to seawater, and check all units and terminology.

Table 2
In the legend, please provide full genus name for all sponges.

Figure 2
Change to “(percent of total)”. Change water to seawater. Also please provide full genus names for all sponges.

Supplementary Figure 1
I think it would be more appropriate to have the same polymer type together so it is easier to see compare the control vs treatment. For example,

A a B b
C c D d
E e F f
G g H H
I i J j

Also, it would be useful to add titles above the 4 columns.
Water without heat Bleach and heat Water without heat Bleach and heat

Experimental design

General concept and experimental design:

A) Sponge morphology

Line 207: this worries me somewhat. While I understand that you are interested in what is retained within the tissue, the very nature of sponge oscula is that they operate as exhalant pores discharging water and items from the spongocoel to the outside – so you may potentially be washing away particles that the sponge is actively discharging at the time of collection. Perhaps a future experiment could focus on collecting this discharge and looking for MP.

Line 443 mentions differences in PMP concentration across the species but there is no in depth discussion (other than for fibres) of the size classes and how they relate to ostia size of each species. Please consider the morphology of the six sponge species investigated (i.e., size of oscula between species) and how this compares to the PMP loads found within them.

Following on from this…
Line 262: Particles are sorted into 9 size ranges. Please provide an explanation as to why these categories are important – do they correspond to the diameter of osita/oscula in the various species investigated?

B) Background contamination

Line 216: not pre-filtering solutions prior to use, especially household bleach that likely comes in a plastic container (the authors do not provide any detail) is a significant flaw in this study. Even though the authors go to great lengths to state this limitation upfront and in their results section (line 334-338), I have very little confidence in the results.

Line 520: your results show it is imperative that filtering of ALL solutions be done, otherwise you cannot trust your results, which unfortunately is the outcome of this study.
To overcome this a net correction was done. Should you have used a chemical method such as Raman or FTIR, you would have been able to identify the polymer type and only exclude those that were found in both blank and sample (see F. Kroon, C. Motti, S. Talbot, P. Sobral, M. Puotinen, A workflow for improving estimates of microplastic contamination in marine waters: A case study from North-Western Australia, Environmental Pollution, 238 (2018) 26-38.). I am presuming that access to such equipment was not possible at the time this work was conducted, in which case the authors need to include:
1) a statement in the introduction or methods regarding the choice of technique, i.e., as a first screen of samples to identify PMP for further analysis,
2) a statement in the discussion strongly recommending that FTIR or Raman be used for future studies to confirm the findings presented here. If the samples are still available it would be worth analysing them by one of these methods and comparing the PMP found here with those confirmed to be MP by chemical analysis.

C) Polymer characterisation

Fluorescence microscopy, especially how it is applied here, has limited capacity to establish an item as a polymer. It can help to detect MP within a sample (i.e., to screen samples and identify those that should be further characterised) but cannot identify the type of plastic polymer. Techniques such as Raman, FTIR, MS are needed to confirm chemical composition. The authors need to state this more strongly. Results reported on Line 305 highlight this, with some polymers able to be detected based on their fluorescence, while others not. The authors show their method of choice is limited – but offer no alternative method of verification.

Line 508: the authors again state the limitations of this fluorescence method to detect MP, and mention brightfield microscopy – was this method used by the authors as a secondary method of PMP identification?

Also please read Monteleone et al 2021 https://doi.org/10.1016/j.jece.2020.104769

Validity of the findings

I have reservations about the PMP loadings found, and specifically the emphasis placed on them likely being MP, given they did not filter the bleach before using. The authors are transparent in this, and with hindsight I have no doubt pre-filtering of ALL solvents used will be incorporated into any future study they undertake (a lesson learned). However, given the plethora of studies now published that have focused on improving workflow practices and eliminating or accounting for background contamination, the lack of implementation of pre-filtering is of concern, i.e., suggests the authors were not familiar with the literature BEFORE conducting the experiment.
The authors need to address this and the other issues I've raised.

Additional comments

Comments on the text

Abstract:
Line 46: There is no information provided in the abstract that indicates the 'resistance'. Please reconsider this phrase or delete.

Introduction:

Line 56: Please be careful here, nutrients and sediments are natural. If you are referring to land runoff and fertilizers then state that explicitly, i.e., “The detrimental effects of anthropogenic pollutants such as chemicals (provide examples in brackets – wastewater discharge, oil spills, etc), nutrients from fertilizers, and sediment from land runoff on the environment are well known…”

Line 57: the release of microplastics IS of great concern – we are currently worried, so be stronger in this statement. For example “but the rapid proliferation of microplastic (MP) contamination is now an issue of increasing concern”

Line 64: change degradation to fragmentation.

Line 75: change to “the overall impact of MP”. The use of ‘role’ suggests it is part of the system – MP are NOT natural and do not belong, so they impact.
Line 78-82: please place each citation after the organism type – some readers may be interested in a specific organism and would appreciate being directed to the appropriate citation.

Line 108: change to “The removal of these food particles from the water column by sponges…”

Line 112: Please elaborate on the ‘widespread distribution’. Sponges have a broad spatial and temporal distribution, with many varied habitats, i.e. from mangroves, tidal pools, coral reefs, CO2 seeps, deep water, cold water….

Line 157: write ‘as per Covernton’. Also by your definition items are not MP but PMP so please modify this sentence. I suggest “We used fluorescence microscopy to identify and quantify potential MP (PMP) as per Covernton et al. (2019).”

Materials & Methods:

Line 185: Please change one-liter to 1 L (same for Line 373). Also how were the 1 L glass jars cleaned? It is important to provide such details, especially as background contamination is such an issue in this type of research.

Line 197: please explain what analytical grade water is – reverse osmosis? Milli-Q? Purchased from a supplier? Has it been filtered, if show what range filter size? Same comment applies to Line 281.

Line 201: Suggest you change this to “One procedural blank was run in parallel to each of the three seawater samples”

Line 215: change ‘shows’ to ‘causes’

Line 219: change to “to evaluate the degree of pre-existing experimental contamination.”

Line 219: Please change to “Vials with sponge tissue were heated to 60°C for 2 hr to expedite digestion.” Also, please provide references to support the use of bleach and heating to recover MP. Bleach is a strong oxidising agent and can damage some plastics, heat is known to exacerbate (or accelerate) chemical reactions. And include reference to the section “Positive Controls” specifically instead of saying (see below).

Line 240: the term “others” is used – please consider a more appropriate name. For example, ‘unknown polymer’ (Punknown). “Others” is very unscientific.

Line 243: Please rephrase this – it is clumsy. I suggest “Two controls (without MP added) were also included. The first was bleach digestion (X mL) with heat (N=10) and replicates were processed according to the sponge sample procedure. The second was MilliQ (X mL) without heat (N=10).” Its not clear to me how these were then processed – by adding an additional 5-10 mL? or simply filtered?

Line 253-254: this sentence reads like a result. I suggest rephrasing and providing relevant citations. For example:
“Potential MP was distinguished from fluorescing background material based on the brightness and color of fluorescence. Inorganic sand grains, proteinaceous spongin, invertebrate cuticle fragments, etc, generally emit a dull, blue-green fluorescence while many plastics fluoresce stronger and with an electric blue color (Figs. 2, S1) (citations)”.

Line 255-258: these two sentences seem to be saying the same thing. Please clarify. If they are a duplication then please combine, if not then please provide further explanation. Also were PMP assessed using standard stereomicroscopy (to determine true color, and also texture)? If so, then details of this needs to be included here. Visual assessment as a component of MP characterisation is widely used –

For visual assessment see F. Noren, Small plastic particles in Swedish West Coast waters, N-research, Internasjonale Miljøorganisasjon (KIMO), Lysekil, Sweden, 2007

For visual assessment within a workflow see F. Kroon, C. Motti, S. Talbot, P. Sobral, M. Puotinen, A workflow for improving estimates of microplastic contamination in marine waters: A case study from North-Western Australia, Environmental Pollution, 238 (2018) 26-38.

Line 288-291: this is repeating information from line 212. Please delete from here and combine with line 212. There is no need to remind the reader.
For line 212 I suggest: “Each subsample was cut into pieces with a utility blade and added to a clean 20 ml glass scintillation vial and covered with foil. In total, 54 subsamples were generated (6 species × 3 replicates × 3 subsamples).”
And then for Data analysis I suggest “The blank-corrected number of PMP g-1 was determined for each of the 54 subsamples representing the six species”.

Line 295: please cite R correctly, including the version used. Refer to https://rdrr.io/r/utils/citation.html

Line 305: “PETE, Polyester” Polyester is PETE!

Results:

Line 310: Please change to “Exposure to bleach and heat did not impact on plastic fluorescence behavior (Fig. S1).”

Line 315: please change to “comprised ~25% of…”, similarly for line 316 “, with ~27% of all particles…”

Line 332: please change to “(17 of the 54…”

Line 340-343: these sentences are confusing. You state that “Most PMP found in all sponge samples…” Yet in the preceding paragraph you stated that 17 subsamples returned a -ve net value, so not every sponge had PMP present. Please clarify.
I presume you mean: “PMP found in sponge subsamples and blanks were mostly very small (10-20 m), comprising nearly half of the total number of PMP detected. These were mostly found in A. cauliformis, A. compressa, C. vaginalis, M. laevis and the blanks, while only 32% and 25% of very small particles were found in I. campana and N. erecta, respectively (Fig. 3).”

Line 343: Please change to “Together, very large particles, small fibers (501–1000 μm) and medium fibers (1001–3000 μm) comprised a large percent (~25–31%) of the total number of particles in I. campana, M. laevis, N. erecta, but not in the other species or blanks (Fig. 3).”

Line 347: please change to “were found only in sponge samples….”

Discussion:

Line 355: Please change to “These previous studies…”

Line 374 and Line 383: Line 383 repeats information on line 374. Please revise so there is minimal repetition. I suggest for Line 383 “Covernton et al (2019) reported 5.28 MP L-1 from 8-m filtered surface seawater samples in coastal British Columbia”.

Line 379: You state “The authors highlighted the necessity of using bulk seawater samples and sensitive filtration methods (ability to detect plastics down to 10 μm) when assessing the exposure of marine organisms to MP pollution (Covernton et al., 2019).” Terminology is very important here. Sensitive filtration methods do NOT give the ability to DETECT plastics, rather to collect MP down to 10 m. Only chemical techniques can detect plastics.

Line 387: Please change to “Although our subsurface value of 107 PMP L-1 corroborates Norén & Naustvoll (2010), it is elevated compared to the other studies.”

Line 398: change to “indicate that they do ingest MP”

Line 399: This is a little confusing. I presume you mean “Raman spectroscopy of items recovered from tropical sponges of Indonesia, digested with bleach and filtered over 1 m membranes, revealed the presence of naturally occurring items such as minerals and shell fragments, as well as anthropogenic items (91-612 PMP g-1 dry tissue; 5-200 m) including cotton and polystyrene (Girard 2020). Of note was one sponge, Ircinia sp., which was found to have 159 polystyrene MP g-1 dry tissue.”

Line 423: change to “within the bodies of the six sponge species…”

Line 440: There is a wealth of information available regarding sponge egestion of particles (shells, minerals of similar size range to MP) – surely this information could inform the potential of your species to egest the MPs, even though MP egestion itself may not have been investigated.

Line 451: agree that particle spatial variation is an important factor, however, so is the preferred size spectrum of the sponge.

Line 457: agree, but should also investigate relationships between these six species from sites that are considered to be more and less contaminated with MP. That would really confirm that they are simply filtering and not selecting for MP.

Line 468: this statement is only true if the gastrointestinal tract and contents is consumed. There is next to no evidence of plastic contamination in flesh, although there is evidence of other chemicals such as plasticisers in flesh. Please provide some detail to support this statement, or if the GIT is not eaten, then better to simply remove any reference to seafood consumption.
I suggest “…susceptible to MP ingestion. With entry of MP into local waterways expected to increase as tourism and residency rates continue to increase, this risk to marine species will be exacerbated (refs).”

Line 483: Please change to “These values are far greater than the 8,000 PMP we predict to be present in a sponge at any given moment in time.”

Line 488: please change to “we did not detect any external effect…”

Line 490: please change to “of effect, with MP exposure having little impact”

Line 492: yes, it maybe that they can resist MP ingestion, or it maybe that they have the ability to increase the egestion rate if the particle load within the spongocoel reaches a threshold. See McGrath 2017 https://doi.org/10.1016/j.jembe.2017.07.013. Note however, there is a threshold of sediment load after which the sponge is likely to not be able to egest, and health begins to suffer. See Pineda 2017 https://www.nature.com/articles/s41598-017-05251-x.

Line 497: please change to “resistant specifically to MP exposure…”

Line 514: given you are likely underestimating the MP presence you cant really say that sponges are resistant to MP uptake – you need to state this again.

Line 537: bleach and heating is known to affect some plastics – just look up any on-line compatibility chart. Also, please note that the extent of the effect was assessed by fluorescence microscopy only – it may be that the polymer integrity is impacted on a chemical level.

---

## Round 0.2 · Minor Revisions

I was able to get a second review from Reviewer 1, who recommends additional edits, mostly related to overstated conclusions based on the limitations of the methods. I agree with these suggestions and ask that you return a minor revision that addresses these comments.

Reviewer 1 ·

Basic reporting

I commend the author on providing a thorough and detailed review of the literature and providing pertinent background information. On second review, I noticed a few instances of providing almost too much detail, superfluous language, and over-use of parentheses in some cases.

Line 34: starting with “This is surprising…” could be reconfigured to avoid use of that type of terminology

After defining PMP, refer to all following instances with the same. Line 40 says ‘to retain any MP present” and lines 42 and 165 uses “potential MP”

Line 98: “Prolific” here can be eliminated. Just state that sponges often exhibit high pumping rates

Line 136: Delete “also”

Lines 152-153: Not sure it is necessary to include detail on symbiont derived nutrition

Lines 177-178: Exclude “also” from both sentences

Lines 245-146: Do you need the parentheses?

Lines 281-282: I don’t think the parenthetical information is important to include “(the length of the 10x10 grid was equal to 1 mm…)”

Lines 337: You used a different spelling of fluorescent here compared to elsewhere in paper.

Line 348: Reconfigure sentence to avoid use of parentheses.

Lines 359-361: Reconfigure sentence to avoid use of parentheses.

Line 383-384: Try to avoid parentheses here too

Line 454: replace method with mechanism

Line 526: “Elevated” is an odd word choice. Elevated as compared to what?

Lines 528-529: Could be abbreviated to “and other coral reef associated marine invertebrates”

Experimental design

The overall description of the methods has been improved. Here are my additional comments:

Methods, data analysis: Why weren’t the water filters blank corrected?

Methods data analysis: Lines 319-321 are confusing. From what I understand, for each species, you collected samples from 3 individuals. Then you broke each individual sample into 3 vials for digestion. Wouldn’t you add the counts from the 3 vials as they are part of one sample, rather than taking the average?

Validity of the findings

While you are, again, very transparent about the limitations of your methodology, I'm not sure this is reflected properly in the extrapolation of results. In general, I would edit the discussion to be even more cautious with your assumptions. Examples include:

Lines 276-280: The variation in counts between different surveys is concerning and doesn’t instill confidence in an accurate count in seawater

Lines 425-429: My biggest concern with this paper is the broad interpretation of the results. Such as in this case, you first you state that your study suggests that sponges ingest MP. Then the next sentence says that you can only tentatively report potential MP in sponge samples, making the first sentence an overstatement.

Lines 535-549: This paragraph is an extrapolation built upon already provisional results. I don’t think it should be included.

Lines 574-575; 579-580: Make it clear that while your methods may miss certain plastic types, they also may be positively skewed by non-plastic fluorescent particles.

Line 617: “Our results further indicate that sponges do ingest MP” ignores the “potential” part of PMP. Rather, they ingest fluorescent particles that resemble MPs

---

## Round 0.3 · accepted · Accept

The authors have addressed the reviewer comments on the revised ms in a satisfactory manner, and it is now recommended for publication.